# Industry-Fit AI Usage for Crack Detection in Ground Steel

**Daniel Soukup** [1,*,†] 🆔**, Christian Kapeller** [1,†] 🆔**, Bernhard Raml** [2] 🆔 **and Johannes Ruisz** [1]

1   Center for Vision Automation and Control, AIT Austrian Institute of Technology, Giefinggasse 4,
    1210 Vienna, Austria
2   Department of Geodesy and Geoinformation, Technische Universität Wien, Wiedner Hauptstraße 8-10,
    1040 Vienna, Austria
*   Correspondence: daniel.soukup@ait.ac.at
†   These authors contributed equally to this work.

**Abstract:** We investigated optimal implementation strategies for industrial inspection systems aiming to detect cracks on ground steel billets' surfaces by combining state-of-the-art AI-based methods and classical computational imaging techniques. In 2D texture images, the interesting patterns of surface irregularities are often surrounded by visual clutter, which is to be ignored, e.g., grinding patterns. Even neural networks struggle to reliably distinguish between actual surface disruptions and irrelevant background patterns. Consequently, the image acquisition procedure already has to be optimised to the specific application. In our case, we use photometric stereo (PS) imaging to generate 3D surface models of steel billets using multiple illumination units. However, we demonstrate that the neural networks, especially in high-speed scenarios, still suffer from recognition deficiencies when using raw photometric stereo camera data, and are unable to generalise to new billets and image acquisition conditions. Only the additional application of adequate state-of-the-art image processing algorithms guarantees the best results in both aspects. The neural networks benefit when appropriate image acquisition methods together with image processing algorithms emphasise relevant surface structures and reduce overall pattern variation. Our proposed combined strategy shows a 9.25% better detection rate on validation data and is 14.7% better on test data, displaying the best generalisation.

**Keywords:** deep learning; machine vision; computational imaging; photometric stereo; industrial quality inspection

## 1. Introduction

The production of steel fit for the creation of high-quality end-products, e.g., wire, is a complex process that involves metallurgical preparation of the raw steel, casting, remelting, rolling, forging, and subsequent heat treatment of the material into intermediate billets (Figure 1). To ensure the best possible results, billets need to be treated, for example, by grinding away cracks at their surface to ensure a flawless surface for further forming steps, e.g., [1]. It is well understood that the early rejection or repair of flawed parts limits production costs and energy consumption [2,3].

This work focuses on optimizing the grinding procedure of billets hot rolled on a roughing mill. Grinding away surface defects is necessary, but there is a trade-off between improving the billets' quality and milling off valuable material. Thus, targeted, reliable, and fast determination of the product's quality condition is required.

In practice, visual quality inspection plays a crucial role for identifying cracks, a task that is often performed by human experts. Often this process is quite time-consuming, because the operator has to frequently switch stations, resulting in a significant bottleneck in the whole steel production pipeline. The goal of our work is to automate the visual crack inspection procedure to support the human mill-operator by presenting automatically generated crack maps for steel surfaces.

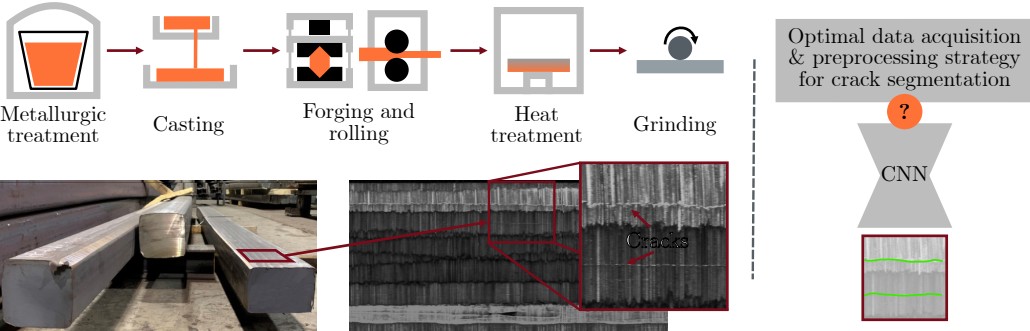

**Figure 1.** Simplified Illustration of a steel production process. (**Left/top**): Process from raw material to intermediate steel fit for processing into further steel products. (**Left/bottom**): Illustration of flawed steel billets and a zoom onto their ground surface, showing grinding patterns (vertical) and cracks on the surface (horizontal). Note that the surface also contains horizontal structures that arise from the grinding process itself that are not considered as flaws in the material. (**Right**): We aim to identify an optimal data acquisition and preprocessing strategy for segmenting cracks in ground steel with a neural network (CNN), especially for the case of time-critical industrial inline inspection.

A visual assistance system that depicts the surface structures of steel parts and highlights potential cracks on a screen in the operators' control station allows them to quickly gain an impression of the surface qualities without having to leave their station. This opens the potential for savings in time and energy, increasing the sustainability of the whole production. Additionally, generated crack maps provide documentation of the surface quality of each produced part, which enables long-term monitoring and quality assurance.

We do not aim to fully automate this process, because there is insufficient and potentially biased data for algorithmic development and training. Our algorithmic image evaluation module involves modern state-of-the-art AI methods (i.e., neural networks for image analysis), which are very dependent on representative data. Hence, human oversight is vital to develop and maintain human trust in the system's predictions, a position that is also understood by the European Commission [4].

Furthermore, it is well known that more reliable detection results can be achieved using a human–machine collaboration [5–8].

By means of this real-world industrial crack detection problem, we demonstrate that current state-of-the-art AI methods for visual quality inspection are optimally introduced in the industry when they go hand in hand with conventional high-quality state-of-the-art machine vision and computational imaging methods. We will demonstrate that common limits in industrial inline inspection, such as training data availability and tight speed requirements, make it necessary to substantiate the information content of the image data by making use of physical/mathematical modelling—already at data acquisition time but also in the algorithmic data evaluation. This helps to suppress disturbing irrelevant surface structures and supports AI methods to identify relevant structures that are more target-oriented.

The novelty of our manuscript consists of the demonstration and analysis of optimal data pre-processing strategies for convolutional neural networks (CNN) in a time-critical industrial defect detection task, where acquisition speed as well as processing speed are crucial (cf. Figure 1). Due to the speed restrictions, image quality has to be sacrificed by minimizing exposure times, as well as the number of photometric illumination units. Our results reveal that a data pre-processing strategy using the photometric stereo algorithm allows for better CNN predictions. Specifically, we will demonstrate that

- Cracks in the presented metal surfaces can be detected more reliably using data acquired with a specialised 3D machine vision setup rather than simple 2D texture images, and

- Calculating surface models from the raw camera data by means of conventional computational image processing algorithms and using those as input for neural network crack detectors rather than raw camera images yields
  - Better crack detection performance and
  - Better generalisation with regards to newly presented objects (i.e., billet surfaces) as well as different image acquisition conditions (lab vs. on-site acquisition).

In a nutshell, modern AI methods, while capable of impressively disentangling complex patterns in large and high-dimensional image data, still fall short when the image data presented contains biased information content. To achieve optimal performance, additional physical/mathematical knowledge should be introduced into the processing pipeline.

## 2. Inline Steel Surface Acquisition Using Photometric Stereo

In the following subsections, we discuss relevant literature regarding photometric stereo-based acquisition methods and elaborate on the design choices of our acquisition setup.

### 2.1. Related Works

Photometric stereo (PS) [9] is a shape-from-reflectance method that allows us to obtain the surface of an object in terms of a normal vector field by illuminating it from several, usually three or more, directions. By assuming a Lambertian surface, that is, the observed light intensity reflected by an object decreases linearly with an increasing angle between camera object and light source, a detailed surface model can be efficiently obtained. In contrast to shape reconstruction methods that rely on texture, such as stereo-matching, PS can obtain good results even for untextured surfaces. Recently, near-field photometric stereo methods (nPS) have gained significant attention, e.g., [10–12]. By thorough modelling of camera and light sources, nPS methods achieve metric 3D reconstruction from multiple object illuminations. In this work, we refrain from employing nPS methods, because they require significant angle-variation of incident light rays and work best when the optical system is in object vicinity (i.e., wide angle lens and nearly placed light sources) [12]. Our system, however, is designed to operate at a significant object distance ($\geq$400 mm), due to safety aspects. PS is a well-known for method for detecting cracks on metallic surfaces [13]. Several approaches have been proposed, including spectral [14] as well as spatial multiplexing [15]. From a theoretical perspective, light sources are optimally arranged at an equidistant azimuth angle at a polar angle of approximately 56 deg [16]. These observations, however, heavily rely on Lambertian surface response, and are not true in general, especially for glossy metallic surfaces. Several proposals address this issue [17,18]. A common theme for obtaining accurate surface models is the use of a high number (e.g., up to 32) of light sources [17,18].

### 2.2. System Design

The design of our photometric stereo acquisition method is guided by the need for (1) high data throughput, (2) the appearance of the observed steel billets, and (3) the necessity of robust crack detection for several steel alloys. The throughput requirement (1) stems from the size of the observed steel billets that have dimensions of 6000 $\times$ 200 mm. Cracks are reliably observable at a resolution of 50 μm/px resolution, corresponding to 120,000 $\times$ 4000 pixels. At a transport speed of 500 mm/s the system has to process 240 Mpx/s. These specifications necessitate to minimise the number of employed light sources, as each additional light source increases the system's throughput. We settled for 4 light sources, as surface normals of acceptable variance can be obtained from the acquisition of three light sources, while the gains in normal variance diminish sharply with each additional light source [16].

The observed surfaces (2) exhibit pronounced grinding structures along their width (see Figure 2), i.e., orthogonal to the transport direction (y-axis in images), while the sought after cracks are aligned in transport direction (x-axis in images). In order to maximise

the signal of the cracks, we place the light sources co-linearly orthogonal to the transport direction. This effectively eliminates the ability of the PS algorithm to estimate normal vectors' slant in y-direction, as can be observed in Figure 2. Moreover, regarding the x-direction, the estimation is based on four different slant angles, as opposed to two slant angles in a system in a standard 2D light configuration [16].

The system should be robust for various steel alloys (3), each of which have different surface properties that are not reliably specifiable a priori. To this end, we settle on the Lambertian assumption, well knowing that it is only a coarse simplification of the real underlying, but unknown, reflectance models. However, we aim to merely detect cracks that appear in the data as distinct discontinuities. We will demonstrate that, for the sole purpose of crack detection, and given our tight speed requirement, the Lambertian assumption is a sufficient and pragmatic choice as a reflectance model.

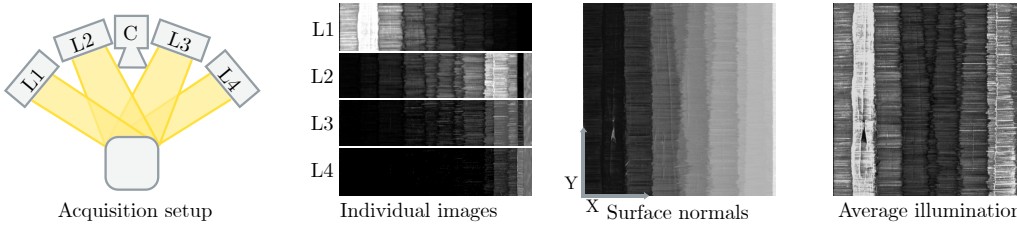

**Figure 2.** Acquisition setup for photometric stereo surface data. Using a setup consisting of a camera (C) and four line-light sources (L1, ..., L4) arranged perpendicular to the steel billet's transport direction (**left**), image data of individual light sources is acquired (**middle-left**). Using the photometric stereo algorithm, we obtain surface normals (**middle-right**). Compared to averaged image data obtained from the same setup (**right**), we observe substantially diminished horizontal grinding patterns.

### 2.3. From Photometric Stereo Images to Surface Structure

The photometric stereo (PS) acquisition setup described in Section 2 yields a stack of images for each region of the acquired object, with one image for each illumination unit. From the varying shading reflected from each object point due to the different illumination angles, the surface orientation of the respective object point can be derived mathematically, i.e., the photometric stereo algorithm [9]. If we assume a Lambertian object irradiation response that is perfect diffuse material illuminated by infinitely distant, parallel light rays, we can describe image formation as

$$I_{i,j} = \rho_{i,j} L n_{i,j}, \tag{1}$$

where $L \in \mathbb{R}^{n \times 3}$ is the illumination matrix containing vectors corresponding to the known illumination directions, relative to the camera centre projected onto the object plane. $I_{i,j} \in \mathbb{R}^n$ are the observed light intensities in pixel position $i, j$ according to the $n$ illumination units, $\rho_{i,j} \in \mathbb{R}$ is the albedo value, and $n_{i,j} \in \mathbb{R}^3$ is the normalised surface normal vector, i.e., the surface orientation in each object point. A 2D map of all vectors $n_{i,j}$ form a model of the sought-after surface structure corresponding to the measured shading variations.

Mathematically, the inversion of $L$ yields the surface normal vector at pixel position $i, j$ scaled by the albedo:

$$\rho_{i,j} n_{i,j} = L^+ o_{i,j}, \tag{2}$$

where $L^+ \in \mathbb{R}^{3 \times n}$ denotes the Moore-Penrose pseudo inverse of $L$.

Due to the special configuration of our illumination setup, where the individual illumination units are arranged along only one dimension (c.f. Section 2) to emphasise the detection of vertical cracks (c.f. y-axis in Figure 2), we only focus on the x-component of $n_{i,j}$ as the relevant feature of surface structures. The estimated y-component would be an unreliable estimate in that lighting configuration anyway. When going from left to

right through the resulting 2*D* maps of all normal vectors or their according *x*-components, respectively, a sudden increase in that *x*-component, followed by a sudden decrease, indicates the presence of a crack (c.f. surface normals in Figure 2). For detection of all cracks, we use a neural network (c.f. Section 3.2) that is trained on those 2*D* maps.

## 3. Algorithms for Crack Detection Using PS Data

Tasks such as crack detection in image data is nowadays usually tackled with deep convolutional neural networks (CNN), e.g., [19–21]). The advantages of using photometric stereo image stacks (PIS) as the basis for deep learning models applied to surface inspection, rather than intensity images, has been reported previously [22,23]. Some proposals use PIS as CNN input data to implicitly learn photometric interrelationships inherently contained in the PIS input [18,24,25]. We will demonstrate, however, that using surface structure models obtained by conventional PS (c.f. Section 2.3) as input for a CNN crack classifier yields better detection and generalisation performances. As we will discuss in Section 4.2 in more detail, this becomes especially relevant in industrial use cases, where image quality must be sacrificed due to other conflicting requirements, such as high-speed.

### 3.1. Annotation: Ground-Truth Crack Masks

Annotation masks are images superimposed on the images to be analysed (here: steel surface images), wherein regions of different pattern classes (here: cracks, background) are marked with different colours, e.g., by a human annotator using a drawing software. These masks are used as a ground-truth to train and evaluate CNNs, where the degree of accordance with a detection method's respective prediction masks serves as a measure of detection performance.

For annotation efficiency, we use wide annotation strokes to mark only the rough courses of the cracks (Figure 3). While accepting a certain pixel-wise inaccuracy in the process, the CNN crack detection performance is not impaired in general, as will be shown in Section 4.2.

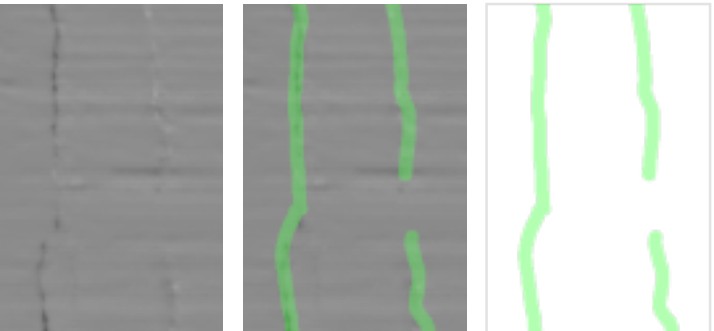

**Figure 3.** Wide annotation strokes mark rough crack course. Annotation is not precisely pixel-accurate. (**Left**): Photometric surface structure model. (**Right**): Crack annotation mask. (**Middle**): Overlay of surface structure model and annotation mask.

### 3.2. Neural Network for Crack Detection in Image Data

We formulate the crack detection problem as a semantic segmentation task, which means, that each pixel is assigned one of the two classes *'crack'* or *'background'*, respectively. An according machine learning model, the so-called *U-Net*, was proposed by Ronneberger et al. [26] for the semantic segmentation of biomedical images and is still often used, e.g., recently, by Buchholz et al. [27], whereas we implemented each convolutional layer as a residual unit [28]. There, patches of images are input into a convolutional neural network (CNN), which learns to output according to prediction masks aligned with those patches. These masks become more and more congruent with correspondingly cutout annotation mask patches (ground-truth) in the course of training the CNN. Thus, the CNN learns to classify each individual pixel in the input images. U-Net comprises an

encoder part, which projects the data to a lower-dimensional feature space, and a decoder part, which decodes those low-dimensional codes into crack prediction masks (c.f. Section 3.1). The low-dimensional link is called the bottleneck-layer. Additionally, so-called skip-connections generate short-cuts between correspondent resolution levels of the encoder and decoder parts.

In their recent paper, Saiz et al. [29] present a system similar to ours for a defect segmentation task on steel components with various coatings. They use a light-dome with 4 light sources, a linear stage that transports the objects in front of a line-scan camera, and they investigate various model architectures including U-Net. They demonstrate that the combination of texture images with photometric stereo images as input for a neural network proposes better prediction performances than pure texture images. However, they use 3 input channels (texture, range, and curvature) and eventually provide segmentation results in a 64 times lower resolution than their input with their best method. In our work, in order to meet processing speed restrictions at the actual production site involving the entire acquisition and processing pipeline, we constrain the input layer to a single channel (see Section 4.1) and provide prediction masks at the highest resolution.

For the semantic segmentation of the cracks on our image data, we apply a U-Net architecture. In our case, we classify pixels as 'crack' or 'background' pixels. A detailed description of the training procedure is provided in Section 4.1, together with an overview of the used training and test data.

## 4. Experiments

To demonstrate the advantage of our proposed method, i.e., using a surface model derived with a conventional photometric stereo as the input for the neural network crack detector, we conduct comparative experiments. We aim to experimentally prove that the approach not only leads to better crack detection performances, but also that CNNs' processing surface model data generalise better to new billet samples, as well as to different acquisition conditions. CNNs that were trained on data acquired under lab conditions perform better on data acquired under on-site conditions, if their inputs are surface normals.

### 4.1. Data Sets and Compared Data Preprocessing Strategies

We compare and analyse the performances of different image preprocessing strategies for training multiple CNNs. 3 experiment configurations using respective *preprocessing strategies* as inputs for those CNNs are considered:

- *'PIS-Raw'*: PIS stack of 4 images corresponding to the 4 illumination units, as they come from the camera. Similar to [25], who use a multi-illumination setup and train a crack classifier on all images without applying the PS algorithm.
- *'PIS-PS'*: Surface model calculated with the photometric stereo algorithm from the PIS, i.e., 2$D$ images/maps, containing the $x$-components of the derived surface normal vectors per object point or pixel, as explained in Section 2.3.
- *'Non-PS'*: Standard gray-scale imaging with a single illumination unit, without taking the photometric stereo into account at all. That strategy was simulated by averaging the 4 images of the PIS.

All image data has been acquired using our acquisition setup consisting of co-linearly arranged illumination units, which has been described in Section 2.

There are 9 *sample objects*, out of which we used 5 small ones ($2000 \times 5800$ px $\approx 100 \times 290$ mm) for *training* the models and *validation* of the training progress (training samples), and 4 larger ones ($2200 \times 90{,}000$ px $\approx 110 \times 4500$ mm) for *testing* the model (test samples). The test samples are full sized steel billets, while the small training samples are cuttings for experimental purposes. The training samples have been acquired in the lab, whereas the large test samples have been acquired with the same acquisition setup on-site at an actual production facility under realistic acquisition conditions, where dust, heat, and vibrations are common and interfere with the acquisition process.

The neural network training, i.e., the actual updates of the CNN weights, is only conducted on the basis of defined training regions on the training samples. Analogously, we define disjoint validation regions, which are only used for measuring the training progress itself. The test data is used to evaluate the detection performances as well as generalisation capabilities of the readily trained CNNs, and are entirely excluded from the training procedure.

We employ 3-fold cross-validation to determine the recognition performances of the CNNs on unseen data. Each training image and its annotation mask are split into 3 equally sized stripes, whereas 2 of those regions are used for training, and the remaining region serves as a validation region. The 3-fold partition allows us to perform cross-validation runs—during each run, another stripe plays the role of the validation set. Hence, each of the 3 experiment configurations is trained 3 times, once for each data partition, resulting in 9 CNNs. To evaluate the performance of each image preprocessing strategy on validation and test data, we average the scores of all 3 cross-validation runs.

Each CNN is trained on random patches of $256 \times 256$ pixels (spatial input size of the CNNs) sampled from training regions of the respective preprocessing strategy data set, together with the corresponding ground-truth masks. We used a batch-size of 10, batch-normalisation [30], an Adam optimizer [31], and trained the CNN for 50 epochs with 100 iterations per epoch. A new set of patches is randomly sampled from all training regions for each epoch. Class occurrences in our data set are unbalanced to a high degree, because there are by far larger regions without a crack than there are regions containing cracks. This is caused by a rather optimised production process. We addressed the class imbalance in the CNN training process by ensuring that half of the patches in a batch contain cracks, resulting in more balanced training batches. Evaluation of validation regions and test samples is performed in a sliding-window manner, covering the entire region to be evaluated.

### 4.2. Results and Discussion

The performances of the described experiment configurations are measured on the basis of the consensus between the ground-truth crack masks and CNN crack prediction masks of corresponding regions.

We use True Positive Rate (TPR, recall) and True Negative Rate (TNR, specificity) over all pixels of evaluated regions as quantitative evaluation metrics:

$$\text{TPR} = \text{TP}/\text{P}, \tag{3}$$

$$\text{TNR} = \text{TN}/\text{N}, \tag{4}$$

where P and N are the counts of defined crack (P—positives) and background (N—negatives) pixels in the ground-truth masks, while TP and TN are the counts of correctly predicted crack and background-pixels in the CNN crack prediction masks. TPR represents the ratio of correctly predicted crack pixels (with regards to the ground-truth), while TNR gives the amount of false alarms in non-crack regions. Although TPR and TNR highlight different individual aspects of classification quality and are valuable to distinguish, we additionally provide their geometric mean as a single unified performance indicator:

$$\bar{\text{G}} := \sqrt{\text{TPR} \cdot \text{TNR}} \tag{5}$$

*Quantitative analysis*. Table 1 shows the performance results (TPR, TNR, $\bar{\text{G}}$) for validation and test data where the results of each experiment configuration (i.e., table line) are averaged values of results from the cross-validated CNNs.

TNR is greater than 99.66 for each experiment configuration and data set, and the mutual differences are very small. This indicates that none of the CNNs generates a significant amount of false alarms, which is advantageous for avoiding operator fatigue.

In terms of the actual crack detection performance, there are indeed distinct deviations between the three preprocessing strategies. PIS-PS yields significantly better TPR than

the other methods, followed by PIS-Raw, that is 40.55 for PIS-PS compared to 31.30 for PIS-RAW. The advantage of PS data as input for CNNs for crack detection is also reflected in corresponding $\bar{G}$ values.

**Table 1.** Performance results for validation data (**left**) and test data (**right**). *Note*: Validation data is generated from the same sample objects as the training data and, like those, acquired in the lab. Test data is generated from different sample objects acquired on-site in the real inspection environment.

| *Valid (Lab)* | $\mathbf{TPR}_{valid}$ | $\mathbf{TNR}_{valid}$ | $\mathbf{\bar{G}}_{valid}$ | *Test (On-Site)* | $\mathbf{TPR}_{test}$ | $\mathbf{TNR}_{test}$ | $\mathbf{\bar{G}}_{test}$ |
|---|---|---|---|---|---|---|---|
| PIS-PS | 40.55 | 99.66 | 63.57 | PIS-PS | 26.83 | 99.58 | 51.69 |
| PIS-Raw | 31.30 | 99.69 | 55.86 | PIS-Raw | 12.13 | 99.68 | 34.77 |
| Non-PS | 19.07 | 99.68 | 43.60 | Non-PS | 11.50 | 99.91 | 33.90 |

Naturally, a certain performance decline from validation data to test data is to be expected. Validation data is drawn from the identical distribution as the training data, in the sense that they both are sampled from (disjoint regions of) the same training objects and have therefore both been acquired under the same friendly conditions in the lab. The test data were derived from totally different steel billets, and their acquisition was conducted on-site at an actual production facility under rough environmental conditions. The lower such a performance decrease is, the better a system is said to generalise from training data to actual real data, i.e., the better is the generalisation capability of the system. Remarkable in this context is the comparable large performance decline of the PIS-Raw preprocessing strategy. While still significantly outperforming the Non-PS data setting on validation data, it is barely ahead (TPR 12.13 vs. 11.50) of that setting on test data.

We take a closer look at the generalisation capabilities of the individual preprocessing strategies by quantitatively comparing their performance declines (Table 2).

**Table 2.** Performance decline, comparing validation data with test data. (**Left**): Absolute decrease ($\Delta_{abs}X := X_{test} - X_{valid}$). (**Right**): Relative decrease ($\Delta_{rel}X := \Delta_{abs}X/X_{valid}$).

| *Abs. Dec.* | $\mathbf{\Delta}_{abs}\mathbf{TPR}$ | $\mathbf{\Delta}_{abs}\mathbf{TNR}$ | $\mathbf{\Delta}_{abs}\mathbf{\bar{G}}$ | *Rel. Dec.* | $\mathbf{\Delta}_{rel}\mathbf{TPR}$ | $\mathbf{\Delta}_{rel}\mathbf{TNR}$ | $\mathbf{\Delta}_{rel}\mathbf{\bar{G}}$ |
|---|---|---|---|---|---|---|---|
| PIS-PS | −13.72 | −0.08 | −11.88 | PIS-PS | −0.34 | 0.00 | −0.19 |
| PIS-Raw | −19.17 | −0.01 | −21.09 | PIS-Raw | −0.61 | 0.00 | −0.38 |
| Non-PS | −7.57 | 0.23 | −9.70 | Non-PS | −0.40 | 0.00 | −0.22 |

Again, TNR does not play a crucial role, with a relative decline close to zero. The performance deviations are primarily reflected in the crack detection performances, i.e., the TPR, which carry over to $\bar{G}$. As already mentioned, PIS-Raw shows the biggest decline. The absolute performance decline (e.g., $\Delta_{abs}\text{TPR} := \text{TPR}_{test} - \text{TPR}_{valid}$) might suggest, at a first glance, that Non-PS is optimal in terms of generalisation, because it shows the smallest decrease. However, Non-PS only yields very poor baseline TPR in validation and test data compared to PIS-PS (c.f. Table 1). Therefore, we rather attribute more relevance to measuring the relative performance decline (e.g., $\Delta_{rel}\text{TPR} := \Delta_{abs}\text{TPR}/\text{TPR}_{valid}$), which also takes the original crack detection performance into account. To illustrate, consider the extreme example of a model which simply classifies every pixel as background. Such a classifier would have the perfect absolute performance decline of 0, but never detect a single crack correctly. The relative measure reveals that PIS-PS not only enables the best crack detection performance, but also generalises the best. In terms of pure generalisation, Non-PS shows passable performance, but the crack detection based on 2D texture images is definitely inadequate with an $\text{TPR}_{valid}$ of 19.07 or an $\text{TPR}_{test}$ of 11.50, respectively. On the other hand, PIS-Raw disqualifies for an actual inspection task due to its very poor generalisation capability ($\text{TPR}_{valid}$ 31.30 vs. $\text{TPR}_{test}$ 12.13).

We ascribe the observed overfitting behaviour for PIS-Raw input data to rather high noise-levels in the raw camera images, which seem to induce a certain kind of bias depending on the actual acquisition condition. Additionally, by only selecting the *x*-component as input in our PIS-PS experiments, we effectively regularise the model to prefer cracks in

transport direction, and remove any noise that would have been introduced by the irrelevant components of the normal vector. It is the challenging speed-limits during inspection that require very small exposure times and therefore lead to rather dark and noisy raw camera images. Image quality of individual camera images must be sacrificed here. The PS algorithm and also the averaging procedure to generate the non-PS 2D texture data (this averaging would also be implicitly performed in a non-simulated real 2D texture image acquisition procedure, because there the exposure time could be chosen to be 4 times longer than in PS acquisition due to only one necessary illumination compared to 4 for PS) cause implicit denoising, which seems to prevent overfitting and enable better generalisation.

*Qualitative analysis*. Statistical evaluation gives an indispensable broad view of detection performances. However, image examples of prediction masks around actual defects provide a visual impression that helps to better rate these values. Figure 4 shows the results for two parallel cracks, where the left one is distinctly visible and the right one only is barely discernible. The distinct one is detected using any of the preprocessing strategies. The thin crack is only detected with the PIS-PS strategy, which demonstrates its higher detection reliability. As our annotation masks only mark the rough courses of the cracks (cf. Section 3.1), they cannot be exactly congruent with the prediction masks, even when the crack instances are indeed detected. Those are accepted pixel-wise inaccuracies to reduce annotation time and therefore costs. As can be observed, the crack instances themselves are detected anyway. That is one reason for the low absolute TPR/TNR values in the statistical evaluation. Those are not expressive in an absolute manner here, but rather for comparative analysis of the different configurations' performances. Other authors [29] reported higher absolute performance numbers for a similar task, with a similar system, but for different inspection objects and defect types. However, their defects are distinctly visible in their images compared to our situation and their annotations seem very accurate in a pixel-precise manner. Therefore, better pixel-wise performance numbers are reasonable compared to our situation, and we were forced to sacrifice annotation precision due to cost efficiency.

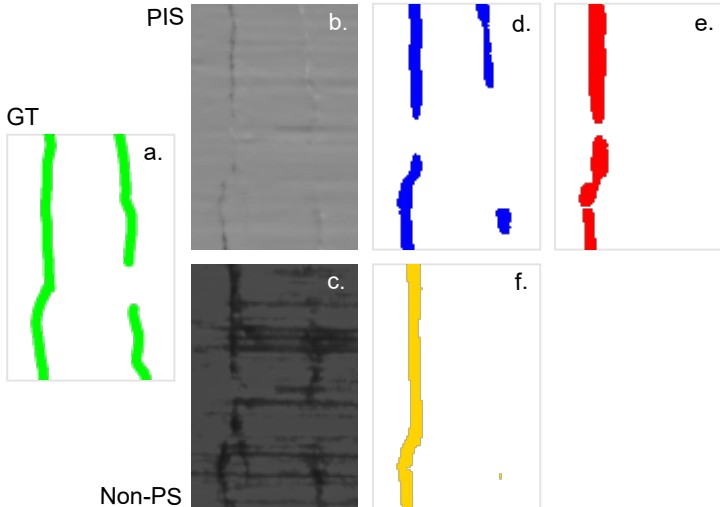

**Figure 4.** Qualitative evaluation based on a 4.25 × 6.25 mm patch. Annotation mask (**a**), PIS-PS surface structure model (**b**), non-PS image (**c**), and prediction masks from CNNs trained on PIS-PS (**d**), PIS-Raw (**e**), and non-PS (**f**) data. The distinct crack left was detected with all three preprocessing strategies, i.e., PIS-PS, PIS-Raw, and Non-PS. The very fine crack right was only detected by PIS-PS, at least partly with regards to the annotation.

Figure 5 depicts a crack, whose visibility changes along its course. While the annotation mark is mainly continuous, the prediction masks' detection strokes comprise piece-wise interruptions. It shows that the neural networks do not mimic the annotation's guidelines blindly, but that they rather only signal a detection if there is actual visual evidence of

a crack. Obviously, the PIS-PS preprocessing strategy is the most adequate to distinctly visualise cracks, as it deviates from the annotation only where the crack's perceptibly is interrupted. Using the PIS-PS preprocessing strategy also enables the detection of tenuous surface disruptions.

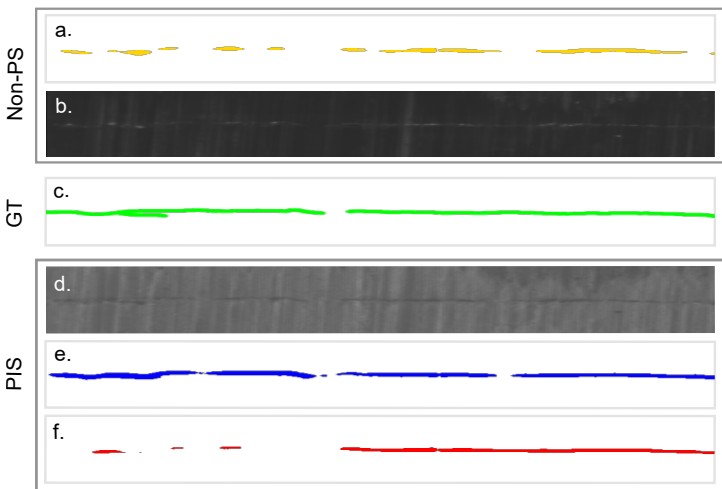

**Figure 5.** Qualitative comparison of tenuous crack detection performance based on a 5.05 × 19.15 mm patch (rotated). Ground-truth annotation mask (middle, **c**), PIS data and results (bottom, **d–f**), and non-PS image and result (top, **a,b**). Only the CNN trained on PIS-PS data detected all perceivable sections of the crack (**e**).

## 5. Conclusions

We have presented a study on the optimal input data content for neural networks trained for the detection of surface disruptions of steel billets. This defect detection step is part of an optical quality inspection pipeline for a real industrial steel billet production process. Our results demonstrate that the distinction of relevant surface irregularities, which are naturally 3D structures, and irrelevant visual structures, such as rust, soil, or grinding patterns, is not reliably feasible on the basis of conventional 2D intensity images. We have demonstrated that an appropriate 3D imaging system based on multi-illumination acquisitions, i.e., photometric stereo, is better suited to visualise the surface structure for the purpose of crack detection, even when neural networks are used as crack detectors. Another challenge is the rather harsh on-site image acquisition conditions at steel production facilities caused by heat, dust, vibrations, and especially very challenging speed limits. Therefore, diminished quality of individual camera images has to be accepted and dealt with. As even neural networks are incapable to sufficiently and reliably disentangle the photometric interrelationships in those raw camera data acquired with our photometric stereo imaging system, the application of adequate state-of-the-art image processing is inevitable to highlight relevant 3D structures in the input data for the neural networks.

We used real steel billet samples that we acquired in our lab as well as at an actual production facility, and conducted experiments comparing two different image acquisition methods as well as three different data preprocessing strategies. Namely, 2D imaging vs. 3D imaging (photometric stereo) were compared. Multiple convolutional neural networks (CNN) were fed with that image data preprocessed in different ways, trained to detect surface cracks, and their detection performances analysed. The photometric data was presented to different CNNs, once directly as raw camera data and once preprocessed with the photometric stereo algorithm, which algorithmically extracts 3D surface structure models from photometric stereo image stacks.

Our experimental results prove that 3D imaging enhances detection performances of surface irregularities on steel billets. However, they also demonstrated that neural networks benefit from adequate state-of-the-art image processing. The implementation

of computational imaging on the raw photometric stereo camera data not only improved crack detection performance, but also resulted in a better generalisation behaviour with regards to new steel billets and different acquisition conditions. CNNs trained with data acquired under friendly lab conditions demonstrated the best inference results on new steel billets acquired under rough on-site conditions only for surface structure models extracted with computational imaging from photometric stereo image stacks. The computational imaging algorithm simplified the task for the neural networks by adding additional model-based information to the raw camera data, and thereby acting as a kind of regulariser by only considering the *x*-component of the surface normals. It reduces noise and eliminates irrelevant texture patterns by fitting a model of photometric interrelationships into the raw camera data. The experiment results clearly demonstrate that neural networks, while in principle being capable of disentangling complex data relationships, still only perform best when they operate on high-quality image data achieved with optimal machine vision acquisition systems and algorithms. This measure reduces distracting pattern variation in the data and already highlights the task-relevant surface properties in the neural networks' input data. An integral approach, where all parts of an image processing pipeline are optimised to the task to be performed, is especially important for industrial imaging applications, where harsh environmental conditions and tight time-limits exacerbate image acquisition, but at the same time still require high detection reliability.

**Author Contributions:** Conceptualization, D.S. and C.K.; data curation, B.R.; methodology, D.S. and C.K.; software, B.R. and J.R.; writing—original draft preparation, D.S., C.K. and B.R.; writing—review and editing, D.S. and C.K. All authors have read and agreed to the published version of the manuscript.

**Funding:** This research received no external funding.

**Acknowledgments:** The authors would like to kindly thank voestalpine BÖHLER Edelstahl GmbH & Co KG, 8605 Kapfenberg, Austria, for permission to use steel billet data presented in this work.

**Conflicts of Interest:** The authors declare no conflict of interest.

## Abbreviations

The following abbreviations are used in this manuscript:

| | |
|---|---|
| PS | Photometric Stereo |
| PIS | Photometric Stereo Image Stack |
| PIS-Raw | Raw Photometric Stereo image stack |
| PIS-PS | 3D surface model extracted from PIS-Raw with PS algorithm |
| Non-PS | Image data not using Photometric Stereo methods |
| nPS | Near-Field Photometric Stereo |
| CNN | Convolutional Neural Network |
| TPR | True Positive Rate |
| TNR | True Negative Rate |
| G-Mean | Geometric mean of TPR and TNR |

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
