# Peer review of "Industry-Fit AI Usage for Crack Detection in Ground Steel"

_electronics, doi:10.3390/electronics11172643_

Round 1

Reviewer 1 Report

 1.     According to the related literature, crack detection using photometric stereo images of metal surface defects and CNN- has been published by several authors [1–3], including semantic segmentation, 3D or 2D images, and other aspects indicated in the manuscript. Therefore, could the authors indicate the novelty of the manuscript?

2.     The authors implement semantic segmentation where each pixel is classified as a crack or background. A similar mechanism is utilized by the transformer architecture, which does not have a good mechanism to capture spatial information inside each patch, i.e., this approach could ignore an important spatial local pattern, such as texture [4]. Could the authors explain this gap?

3.     The manuscript presents a poor review of the works related to photometric stereo images of metal surface defects and CNN.

4.     Equation (4) must be corrected, TNR=TN/N. Why didn´t the authors use the confusion matrix? This has been used by other authors [2] and allows obtaining sensitivity, specificity, accuracy, and precision.

5.    TNR in all instances, whereas TPR40.55 (PI-PIS), 31.30(PIS-Raw),19.07 (Non-PS). These results reveal that the image dataset is imbalanced; there are many methods to address this problem, for example, augmentation technique, transfer learning, or non-supervised training. The objective is to detect cracks; however, the performance achieved is poor compared to other works [2].

6.     The manuscript exhibits many demerits; the revised version must consistently address the questions performed; otherwise, the manuscript will be rejected.  

1.       [1]  Soukup, D.; Huber-Mörk, R. Convolutional Neural Networks for Steel Surface Defect Detection from Photometric Stereo Images. In; 2014; pp. 668–677.

2.        [2] Saiz, F.A.; Barandiaran, I.; Arbelaiz, A.; Graña, M. Photometric Stereo-Based Defect Detection System for Steel Components Manufacturing Using a Deep Segmentation Network. Sensors 2022, 22, 882, doi:10.3390/s22030882.

3.        [3] Logothetis, F.; Budvytis, I.; Mecca, R.; Cipolla, R. A CNN Based Approach for the Near-Field Photometric Stereo Problem. 2020.

4.        [4] Hendria, W.F.; Phan, Q.T.; Adzaka, F.; Jeong, C. Combining Transformer and CNN for Object Detection in UAV Imagery. ICT Express 2021, doi:10.1016/j.icte.2021.12.006.

Reviewer 2 Report

The author tried to develop AI-based methods and classical computational imaging techniques to detect cracks on ground steel billets’ surfaces. The manuscript has been written nicely and all the results are well presented. I have only one concern if the author can make figure 1 a little more interesting.  

Round 2

Reviewer 1 Report

The authors addressed some posed questions; however, one explanation for the poor performance is not reported (main question). Other authors reported better results using a similar approach (see initial review). Therefore, I recommend against publishing this manuscript.
